# Rotavirus susceptibility of antibiotic-treated mice ascribed to diminished expression of interleukin-22

**Daniel Schnepf**[1]*, **Pedro Hernandez**[2], **Tanel Mahlakõiv**[1], **Stefania Crotta**[3], **Meagan E. Sullender**[4], **Stefan T. Peterson**[4], **Annette Ohnemus**[1], **Camille Michiels**[5], **Ian Gentle**[6,7], **Laure Dumoutier**[5], **Celso A. Reis**[8,9], **Andreas Diefenbach**[10,11], **Andreas Wack**[3], **Megan T. Baldridge**[4], **Peter Staeheli**[1,7]*

**1** Institute of Virology, Medical Center University of Freiburg, Freiburg, Germany, **2** Institut Curie, PSL Research University, INSERM U934, CNRS UMR3215, Development and Homeostasis of Mucosal Tissues Group, Paris, France, **3** Immunoregulation Laboratory, The Francis Crick Institute, London, United Kingdom, **4** Department of Medicine, Division of Infectious Diseases, Edison Family Center for Genome Sciences & Systems Biology, Washington University School of Medicine, St. Louis, MO, United States of America, **5** de Duve Institute, Université catholique de Louvain, Brussels, Belgium, **6** Institute of Medical Microbiology and Hygiene, Medical Center University of Freiburg, Freiburg, Germany, **7** Faculty of Medicine, University of Freiburg, Freiburg, Germany, **8** Instituto de Investigação e Inovação em Saúde, University of Porto, Porto, Portugal, **9** Institute of Molecular Pathology and Immunology, University of Porto, Porto, Portugal, **10** Institute of Microbiology, Infectious Diseases and Immunology, Charité - Universitätsmedizin Berlin, Berlin, Germany, **11** Mucosal and Developmental Immunology, Deutsches Rheuma-Forschungszentrum, an Institute of the Leibniz Gemeinschaft, Berlin, Germany

* daniel.schnepf@uniklinik-freiburg.de (DS); peter.staeheli@uniklinik-freiburg.de (PS)

**Data Availability Statement:** All relevant data are within the manuscript. Raw microarray data can be accessed at GEO under accession number GSE166400.

## Abstract

The commensal microbiota regulates susceptibility to enteric pathogens by fine-tuning mucosal innate immune responses, but how susceptibility to enteric viruses is shaped by the microbiota remains incompletely understood. Past reports have indicated that commensal bacteria may either promote or repress rotavirus replication in the small intestine of mice. We now report that rotavirus replicated more efficiently in the intestines of germ-free and antibiotic-treated mice compared to animals with an unmodified microbiota. Antibiotic treatment also facilitated rotavirus replication in type I and type III interferon (IFN) receptor-deficient mice, revealing IFN-independent proviral effects. Expression of interleukin-22 (IL-22) was strongly diminished in the intestine of antibiotic-treated mice. Treatment with exogenous IL-22 blocked rotavirus replication in microbiota-depleted wild-type and *Stat1*<sup>-/-</sup> mice, demonstrating that the antiviral effect of IL-22 in animals with altered microbiome is not dependent on IFN signaling. In antibiotic-treated animals, IL-22-induced a specific set of genes including *Fut2*, encoding fucosyl-transferase 2 that participates in the biosynthesis of fucosylated glycans which can mediate rotavirus binding. Interestingly, IL-22 also blocked rotavirus replication in antibiotic-treated *Fut2*<sup>-/-</sup> mice. Furthermore, IL-22 inhibited rotavirus replication in antibiotic-treated mice lacking key molecules of the necroptosis or pyroptosis pathways of programmed cell death. Taken together, our results demonstrate that IL-22 determines rotavirus susceptibility of antibiotic-treated mice, yet the IL-22-induced effector molecules conferring rotavirus resistance remain elusive.

**Funding:** This work was supported by a grant from the Deutsche Forschungsgemeinschaft to P.S., as well as grants from the NIH (R01 AI141716 and R01 OD024917), the Children's Discovery Institute of Washington University and St. Louis Children's Hospital Interdisciplinary Research Initiative (MI-II-2019-790), and The Mathers Foundation to M.T.B. The funders had no role in study design, data collection and analysis, decision to publish, or preparation of the manuscript.

**Competing interests:** The authors have declared that no competing interests exist.

## Introduction

The microbiota dramatically alters host susceptibility to multiple viral infections via diverse mechanisms [1–4]. Physical interaction with bacteria enhances viral infectivity for some enteric viruses. Binding to bacterial surface lipopolysaccharides (LPS) enhances the stability of poliovirus virions, resulting in decreased susceptibility of mice treated with an antibiotic cocktail compared to mice with undisturbed microbiota [5, 6]. Binding to bacteria further promotes poliovirus attachment to target cells, thereby facilitating simultaneous infection with multiple virions per cell and promoting viral fitness by enhancing the frequency of genetic recombination between different virus strains [7]. Similar mechanisms may account for the observed reduction of reovirus infectivity in antibiotic-treated mice [5]. Reduced vertical transmission of mouse mammary tumor virus through maternal milk in antibiotic-treated mice has also been observed, attributed to a lack of virus-bound bacterial LPS which stimulate the production of the immunosuppressive cytokine IL-10 [8, 9].

For other enteric viruses, both direct and indirect interactions with the microbiota may be at work. Human norovirus infection of B cells was enhanced in the presence of histo-blood group antigen-expressing enteric bacteria [10], and direct binding of human and murine norovirus to commensal bacteria appear to contribute to the proviral effects of the microbiota [11]. Other mechanisms of regulating virus susceptibility rely on modulation of host signaling by the microbiota. Antibiotic treatment decreased replication of persistent murine norovirus in the intestines of wild-type mice but not of mice with defective interferon-λ (IFN-λ) signaling [12], indicating that the bacterial microbiota limits the efficacy of IFN-λ-dependent innate immunity by an unknown mechanism. Interestingly, commensal bacteria were found to facilitate acute murine norovirus infection of distal gut regions while simultaneously inhibiting virus infection of the proximal small intestine [10, 13]. Virus inhibition in the proximal gut is secondary to priming of IFN-λ expression by bacteria-biotransformed bile acids [13].

The microbiota was further shown to restrict rather than to promote viral growth in several other experimental systems [4, 14, 15]. In these cases, commensal-derived signals provided tonic type I IFN-mediated immune stimulation that lowered the activation threshold of the innate immune system required for optimal antiviral immunity. Accordingly, mice treated with an antibiotic cocktail exhibited impaired epithelial, innate and adaptive antiviral responses resulting in delayed viral clearance after exposure to systemic LCMV or mucosal influenza virus infections [4, 15]. Mononuclear phagocytes of microbiome-depleted mice no longer expressed inflammatory response genes so that priming of natural killer cells and antiviral immunity to murine cytometagalovirus was severely compromised [14].

How the microbiota modulates rotavirus replication is incompletely understood, despite this being a pathogen of great clinical importance [16]. Antibiotic-mediated microbiome modulation enhanced fecal shedding of a rotavirus live vaccine in adult volunteers [17], and gram-negative probiotics protected against human rotavirus infection of gnotobiotic pigs [18]. However, antibiotic treatment has been reported to suppress rotavirus infection of gut epithelial cells, with an overall reduction in the incidence and duration of diarrhea in suckling mice [19]. Conversely, some mouse colonies have been found to be highly resistant to rotavirus infection, a resistance that can be transferred to susceptible mice via fecal microbial transplantation [20], indicating that a specific component of the microbiome can mediate antiviral effects. Various probiotics including *Bifidobacterium sp*. have been identified as protective against rotavirus infection [21, 22], and particular segmented filamentous bacteria (SFB) were shown to be sufficient to protect mice against rotavirus infection and associated diarrhea [20]. Interestingly, SFB-mediated protection was independent of known rotavirus-impeding factors such as IFN-λ [23–25], interleukin-18 (IL-18) [26] and IL-22 [26, 27], and instead rotavirus resistance

correlated with accelerated epithelial cell turnover in this experimental system [20]. Bacterial flagellin alone is also sufficient to mediate potent antiviral effects against rotavirus, acting through TLR5 to promote IL-22- and IL-18-dependent antiviral activity [26]. Thus, the relative pro- and antiviral effects of the commensal microbiota on rotavirus infection remain unclear.

IFN-λ and IL-22 are both members of the IL-10 cytokine family. Their specific receptor chains (IFN-λR1 and IL-22Rα, respectively) both associate with IL-10Rβ to form functional heterodimeric receptor complexes, and the *Ifnlr1* and *Il22ra1* genes are close relatives located in adjacent positions in the mouse and human genomes [28, 29]. The IFN-λ and the IL-22 receptors are highly expressed in epithelial cells, and both cytokines cooperate for the induction of IFN-stimulated genes and the control of rotavirus infection in mice [27]. IL-22 dramatically alters epithelial cell expression of a variety of genes, ultimately stimulating proliferation and migration of intestinal epithelial cells toward villus tips, driving increased extrusion of the highly differentiated enterocytes in which rotavirus replicates [26, 30]. However, the antiviral activity of IL-22 is less well-understood compared with its well-described function as a key mediator of anti-bacterial responses [31, 32]. IL-22 expression is controlled by the commensal microbiota [33] and, in turn, IL-22 can modulate the composition of the microbiota [34].

To better understand the role of commensal bacteria in host defense against rotavirus, we performed infections of microbiota-depleted mice and we evaluated the antiviral activity of IL-22 under gnotobiotic conditions. Using antibiotic-treated and germ-free mice we found that the microbiota strongly repressed rotavirus replication in the intestinal tract. Our data indicate that rotavirus susceptibility of microbiota-depleted mice results from poor expression of IL-22 in the intestine. We further show that IL-22 confers rotavirus resistance in microbiota-depleted mice by a mechanism that is independent of IFN-λ or STAT1 signaling, fucosyl-transferase 2, or mediators of inducible cell death.

## Material & methods

### Mice

Conventional specific pathogen free (SPF) C57BL/6J mice of both sexes were purchased from Janvier Labs. Germ-free (GF) C57BL/6 mice were obtained from the Clean Mouse Facility of the University of Bern, Bern, Switzerland. GF mice were kept in autoclaved individually ventilated cages, and sterile water and food was provided. GF mice were handled exclusively in a biosafety cabinet. C57BL/6J mice lacking functional *Fut2*, *Ripk3* or *Casp1/11* genes [35–37] as well as mutant B6.A2G-*Mx1-Ifnlr1*$^{-/-}$ [38], B6.A2G-*Mx1-Ifnlr1*$^{-/-}$*Il22*$^{-/-}$ [27], B6.A2G-*Mx1-Stat1*$^{-/-}$ [39] and corresponding B6.A2G-*Mx1* wild-type mice [38] were bred and housed in the animal facilities of the University Medical Center Freiburg. Other C57BL/6J mice originally purchased from Jackson Laboratories (stock 000664; Jackson Laboratories, Bar Harbor, ME) were bred and housed in animal facilities of Washington University in Saint Louis under specific-pathogen-free (including murine norovirus-free) conditions. Generation of *Ifnlr1*$^{-/-}$ mice was previously described [40]; briefly, these mice were established by interbreeding *Ifnlr1*$^{tm1a(EUCOMM)}$ $^{Wtsi}$ mice and deleter-Cre mice, followed by backcrossing by speed congenics onto a C57BL/6J background. *Ifnar1*$^{-/-}$ (B6.129.Ifnar1tm1) were crossed with these *Ifnlr1*$^{-/-}$ mice to generate *Ifnar1*$^{-/-}$*Ifnlr1*$^{-/-}$ double knockout mice. Sentinel animals of our housing facilities were routinely checked for unwanted pathogens by serological and molecular technologies. These observations yielded no evidence for spontaneous infections of our mice with rotavirus.

### Ethics statement

All experiments with mice were carried out in accordance with the guidelines of the Federation for Laboratory Animal Science Associations and the national animal welfare body.

Experiments were in compliance with the German animal protection law and were approved by the animal welfare committee of the Regierungspräsidium Freiburg (permit G-16/98). All experiments at Washington University were conducted according to regulations stipulated by the Washington University Institutional Animal Care and Use Committee and to animal protocol 20190126, approved by the Washington University Animal Studies Committee.

## Antibiotic treatment of mice

To deplete the commensal microbiota, four- to six-week old mice received drinking water *ad libitum* containing 1 mg/ml Cefoxitin (Santa Cruz Biotechnology, Inc), 1 mg/ml gentamycin sulfate (Sigma-Aldrich), 1 mg/ml metronidazole (Sigma-Aldrich) and 1 mg/ml vancomycin (HEXAL®) for 4 weeks. For experiments at Washington University, six-week old mice received drinking water with 1 mg/ml ampicillin, 1 mg/ml metronidazole, 1 mg/ml neomycin, 0.5 mg/ml vancomycin (Sigma, St. Louis, MO) in 20 mg/ml grape Kool-Aid (Kraft Foods, Northfield, IL), or with Kool-Aid alone, *ad libitum* for 9–14 days. In most experiments we confirmed the absence of live bacteria in fecal samples by plating the material on suitable agar plates.

## Virus stocks and infections

To produce the rotavirus used in Freiburg, six day old suckling BALB/c mice were infected orally with 680 infectious dose 50 ($ID_{50}$) (5 µl of a 1:250 dilution) from a stock containing $3.4 \times 10^7$ $ID_{50}$/ml of murine rotavirus strain EDIM [23]. Four days post-infection, mice were sacrificed and colon samples with content were homogenized twice for 18 sec at 6 m/s using a FastPrep®-12 homogeniser (MP Biomedicals). The homogenate was cleared by centrifugation at 5,000 rpm for 15 min. Supernatants were pooled and filtered using a 10 ml syringe and 4.5 µm sterile filters. The filtrate was aliquoted and stored at -80°C. Viral protein VP6 was quantified by RIDASCREEN® ELISA in comparison to the parental stock. The infectious dose 50 ($ID_{50}$) was determined by orally infecting groups of six C57BL/6 suckling mice with 5 µl samples of 10-fold serial dilutions, ranging from $10^{-3}$ to $10^{-7}$. Mice were sacrificed 24 h post-infection and intestines were analyzed for the presence of RNA encoding for VP6 by RT-qPCR. Adult mice (8–10 week old) were orally infected with $2.4 \times 10^4$ $ID_{50}$ in 100 µl by oral gavage using a gavage needle (19G).

To produce the rotavirus used in Saint Louis, four day old suckling CD-1 mice (Charles River) were infected orally with 400 diarrhea dose 50 (5 µl of a 1:10 dilution in PBS + Evan's blue) from a stock of murine rotavirus strain EC-RV received from Estes Lab at Baylor College of Medicine. Three days post-infection, mice were sacrificed and the entire small intestine and colon with contents were harvested and pooled for weighing. A 20% homogenate was prepared in homogenization media of DMEM (Gibco 11995–040) supplemented with 1% penicillin/streptomycin (Gibco 15140–122) by three rounds of homogenization on ice (homogenize for 1 min, then allow to settle for 30 sec). Homogenate was pooled on ice, aliquoted, and then stored at -80°C until use. The shedding dose 50 ($SD_{50}$) of the stock was determined by orally infecting groups of five adult mice (6–8 weeks old), of both BALB/c and C57BL/6 genotypes, with 100 µl of 10-fold serial dilutions, ranging from $10^{-5}$ to $10^{-9}$ for BALB/c and $10^0$ to $10^{-5}$ for C57BL/6, and analyzing fecal pellets for RV genome copies by RT-qPCR. Adult mice (8–10 weeks old) were orally infected with $10^4$ $SD_{50}$ in 100 µl by oral gavage, preceded by 100 µl of sterile 1.33% sodium bicarbonate solution.

## Viral protein quantification in feces by RIDASCREEN® Rotavirus ELISA

Fecal samples were homogenized according to weight with RIDASCREEN® sample dilution buffer 1 and homogenized twice for 11 sec with 6 m/s using a FastPrep®-12 homogenizer.

Homogenates were cleared by centrifugation [5,000 rpm, 5 min, 4°C] and 100 μl of fecal homogenate supernatants were used for RIDASCREEN® Rotavirus ELISA. ELISA was performed according to the manufacturer's instructions. Optical density was measured at 450 nm wave length by using a Tecan Infinite 200 plate reader.

## RNA isolation and RT-qPCR

RNA was isolated using the Direct-zol™ RNA MiniPrep Kit (Zymo Research). Two-three cm pieces of ilea were homogenized in 1 ml TriFast™ twice for 18 sec with 6 m/s using a FastPrep®-12 homogenizer (MP Biomedicals). Homogenates were centrifuged at 12,000 g for 5 min and supernatant were diluted 1:10 in TriFast™. RNA was then isolated following the manufacturer's instructions. LunaScript™ RT SuperMix Kit (New England Biolabs) was used to generate cDNA from 850 ng of total RNA following the manufacturer's instructions. Resulting cDNA served as template for the amplification of transcripts from *Ubc* (QT00245189, QuantiTect Primer Assay, Qiagen), *Hprt* (mm00446968_m1, Applied Biosystems), *Il22* (forward: 5′- CATGCAGGAGGTGGTACCTT -3′; reverse: 5′- CAGACGCAAGCATTT CTCAG -3′), *Reg3b* (forward: 5′- GCTGGAAGTTGGACACCTCAA -3′; reverse: 5′- GACATAG GGCAACTTCACCTCACA -3′), *Reg3g* (forward: 5′- TTCCTGTCCTCCATGATCAAAA -3′; reverse: 5′- CATCCACCTCTGTTGGGTTCA -3′), *Saa1* (forward: 5′- AAATCAGTG ATGGAAGAGAGGC -3′; reverse: 5′- CAGCACAACCTACTGAGCTA -3′), *Fut2* (forward: 5′- ACCTCCAGCAACGAATAGTGA -3′; reverse: 5′- GCCGATGGAATTGATCGTGAA -3′), *Rps29 (*forward 5′-GCAAATACGGGCTGAACATG-3′; reverse 5′-GTCCAACTTAATGA AGCCTATGTC-3′) by real-time PCR using TaqMan Gene Expression Assays (Applied Biosystems), Universal PCR Master Mix (Applied Biosystems) and the QuantStudio 5 Real-Time PCR system (Applied Biosystems by Thermo Fisher Scientific). Increase in transcript levels were determined by the $2^{-\Delta Ct}$ method relative to expression of the indicated housekeeping genes.

## Intestinal epithelial cell enrichment

Small intestines were collected and cut into 3–6 cm long pieces, mechanically cleaned, rinsed with PBS using a gavage needle (19G) and inverted on customized inoculating loops. Samples were incubated in 30 mM EDTA for 10 min at 37°C. Intestinal epithelia cells were detached from organ pieces by centripetal force using a customary electric drill [approximately 1,000 rpm]. Samples were centrifuged with 2–3 short impulses in 50 ml PBS with 5% FCS. After 1 h of sedimentation, the lower half of the suspension was isolated, cells were pelleted [1,300 rpm, 10 min, 4°C], lyzed in TriFast™ and total RNA isolated using the Direct-zol™ RNA MiniPrep Kit (Zymo Research).

## Transcriptome analysis

RNA for transcriptome analysis was isolated using the RiboPure™ Kit (Invitrogen) according to the manufacturer's instructions. In brief, 2–3 cm pieces of ilea were homogenized in 1 ml TriFast™ twice for 18 sec with 6 m/s using a FastPrep®-12 homogenizer (MP Biomedicals). Homogenates were centrifuged with 12,000 g for 10 min at 4°C, 600 μl of supernatant mixed with 500 μl TriFast™, centrifuged again with 12,000 g for 10 min at 4°C and 1 ml of supernatant was mixed with 100 μl of 1-bromo-3-chloropropane and incubated for 5 min at room temperature. Phase separation was achieved by centrifugation with 12,000 g for 10 min at 4°C. 400 μl of the aqueous phase was then mixed with 200 μl of ethanol and passed through the filter cartridge by centrifugation (12,000 g for 1 min), the filter cartridge was then washed twice with 500 μl wash solution and RNA was finally eluted with 100 μl elution buffer.

## MicroArray

Total RNA harvested from intestinal epithelia cells was hybridized using Illumina Mouse WG-6_V2_0_R0_11278593_A arrays. The raw intensity values for each entity were preprocessed by RMA normalization against the median intensity in mock-treated samples. Using GeneSpring 11.5, all transcripts were then filtered based on flags (present or marginal, in at least 2 out of 10 samples). Moderate t-test (IL22-treated versus mock-treated) was performed to identify genes differentially expressed relative to controls ($\geq$5-fold change; $p < 0.01$, Benjamini-Hochberg multiple test correction). Raw microarray data can be accessed at GEO under accession number GSE166400.

## Statistical analysis

For statistical analyses the GraphPad Prism 8 software (GaphPad Software, USA) was used. Testing for statistical significance was performed on log-transformed values by using ordinary one-way ANOVA with Tukey's multiple comparisons or the unpaired Student's t-test. Asterisks indicate p-values: $^{*}p < 0.05$; $^{**}p < 0.01$; $^{***}p < 0.001$; $^{****}p < 0.001$.

## Results

### Germ-free and antibiotic-treated mice are highly susceptible to oral infection with rotavirus

To investigate the effect of the microbiota on rotavirus susceptibility, we monitored the levels of rotavirus antigen in the feces of three experimentally infected groups of adult C57BL/6 mice, namely conventionally-housed specific pathogen-free (SPF) animals, SPF animals treated with an antibiotic cocktail (Abx), and germ-free (GF) animals. During the 6-day observation period following oral infection with rotavirus, viral antigen was nearly undetectable in fecal samples of SPF mice by ELISA (Fig 1). In contrast, fecal samples from the majority of infected Abx and GF mice contained high levels of rotavirus antigen (Fig 1), indicating that the microbiota has an inhibitory effect on rotavirus replication.

Since an earlier study indicated that the microbiota promotes rather than inhibits rotavirus replication [19], we sought to confirm this finding in an unrelated colony of C57BL/6 wild-type mice at a second independent institution, using an independently prepared rotavirus stock. By measuring viral RNA in fecal samples on day 5 post-infection by RT-qPCR analysis we again found that wild-type (WT) mice with undisturbed microbiota shed only low amounts

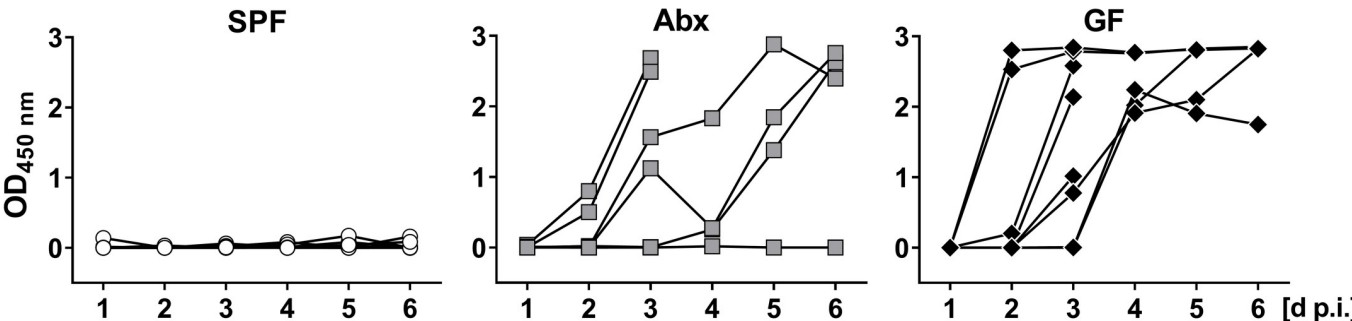

**Fig 1. Germ-free and antibiotic-treated mice are highly susceptible to oral infection with rotavirus.** Conventional specific pathogen free (SPF, n = 10), antibiotic-treated (Abx, n = 9) and germ-free (GF, n = 8) C57BL/6 mice were infected orally with $2.4 \times 10^4$ $ID_{50}$ of murine rotavirus strain EDIM. Fecal pellets of individual mice were collected daily and levels of viral antigen were determined by ELISA. Four animals of the Abx and the GF groups were sacrificed on day 3 post-infection for analysis of tissue in the context of a different project.

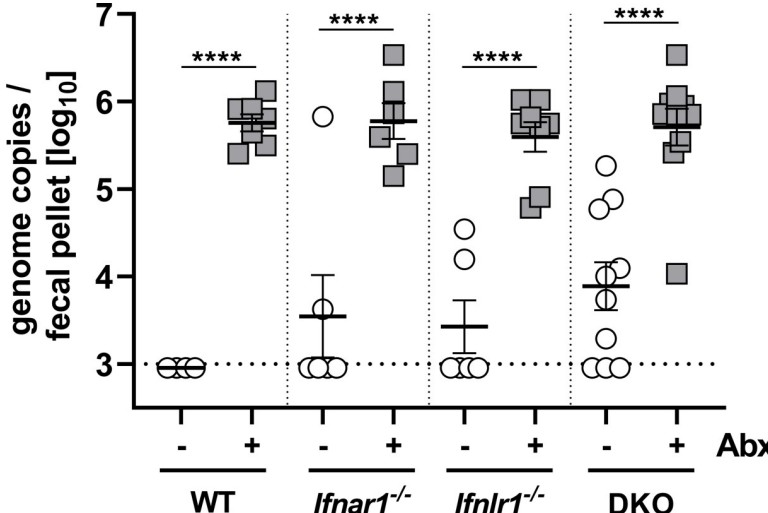

**Fig 2. Rotavirus susceptibility of wild-type and IFN receptor-deficient C57BL/6 mice is greatly enhanced if the microbiome is depleted with antibiotics.** WT (n = 7), *Ifnar1*$^{-/-}$ (n = 6), *Ifnlr1*$^{-/-}$ (n = 8) and *Ifnar1*$^{-/-}$ *Ifnlr1*$^{-/-}$ (DKO, n = 10) mice were left untreated or treated with antibiotics (Abx) before oral infection with $10^4$ $SD_{50}$ of the EC strain of murine rotavirus. On day 5 post-infection, fecal pellets were collected and viral RNA levels were determined by RT-qPCR. Symbols represent individual mice, and bars represent means ± SEM. Line represents limit of detection of the RT-qPCR assay. Statistical analysis: Ordinary one-way ANOVA with Tukey's multiple comparisons; ****p<0.0001.

of virus whereas virus shedding of antibiotic-treated WT mice was strongly enhanced (Fig 2). Prior work indicated that the type I and the type III IFN systems inhibit the replication of rotavirus in the mouse intestine [24, 25]. Therefore, we assessed the possibility that microbiota-mediated rotavirus resistance might depend on IFN signaling. In agreement with earlier studies [24, 25], we observed that animals lacking functional receptors for type I IFN (*Ifnar1*$^{-/-}$), for type III IFN (*Ifnlr1*$^{-/-}$) or for both IFN types (DKO) were more susceptible to rotavirus infection compared with WT mice (Fig 2). Importantly, however, treating these mutant mice with antibiotics also enhanced rotavirus levels on average at least 50-fold (Fig 2), demonstrating that enhanced intestinal replication of rotavirus in mice with reduced microbiota is not only due to impaired activation of type I IFN and/or type III IFN pathways by commensal bacteria. Rotavirus shedding by untreated mutant mice was surprisingly heterogeneous (Fig 2), raising the possibility that the microbiota composition of individual animals could matter. Since our animal facilities are not free of segmented filamentous bacteria (SFB) which are known to confer rotavirus resistance [20], we speculate that the degree of SFB colonization in individual animals could influence intrinsic rotavirus susceptibility.

## IL-22 confers rotavirus resistance in antibiotic-treated mice via a mechanism that is not dependent on IFN-λ or STAT1 signaling

To identify IFN-independent antiviral factors which are present in the intestine of SPF mice but absent or strongly diminished in GF mice, we performed RT-qPCR analysis of cytokines previously linked to rotavirus resistance [24–26]. We found that expression of *Il22* was on average about 200 fold lower in ileum samples from uninfected GF (Fig 3A) or Abx mice (Fig 3B) than in samples from uninfected SPF mice. Furthermore, compared with SPF mice, expression of the IL-22-induced *Reg3b* and *Reg3g* genes was on average 10–20 fold reduced in the ileum of GF (Fig 3A) and Abx mice (Fig 3B), demonstrating that the high rotavirus

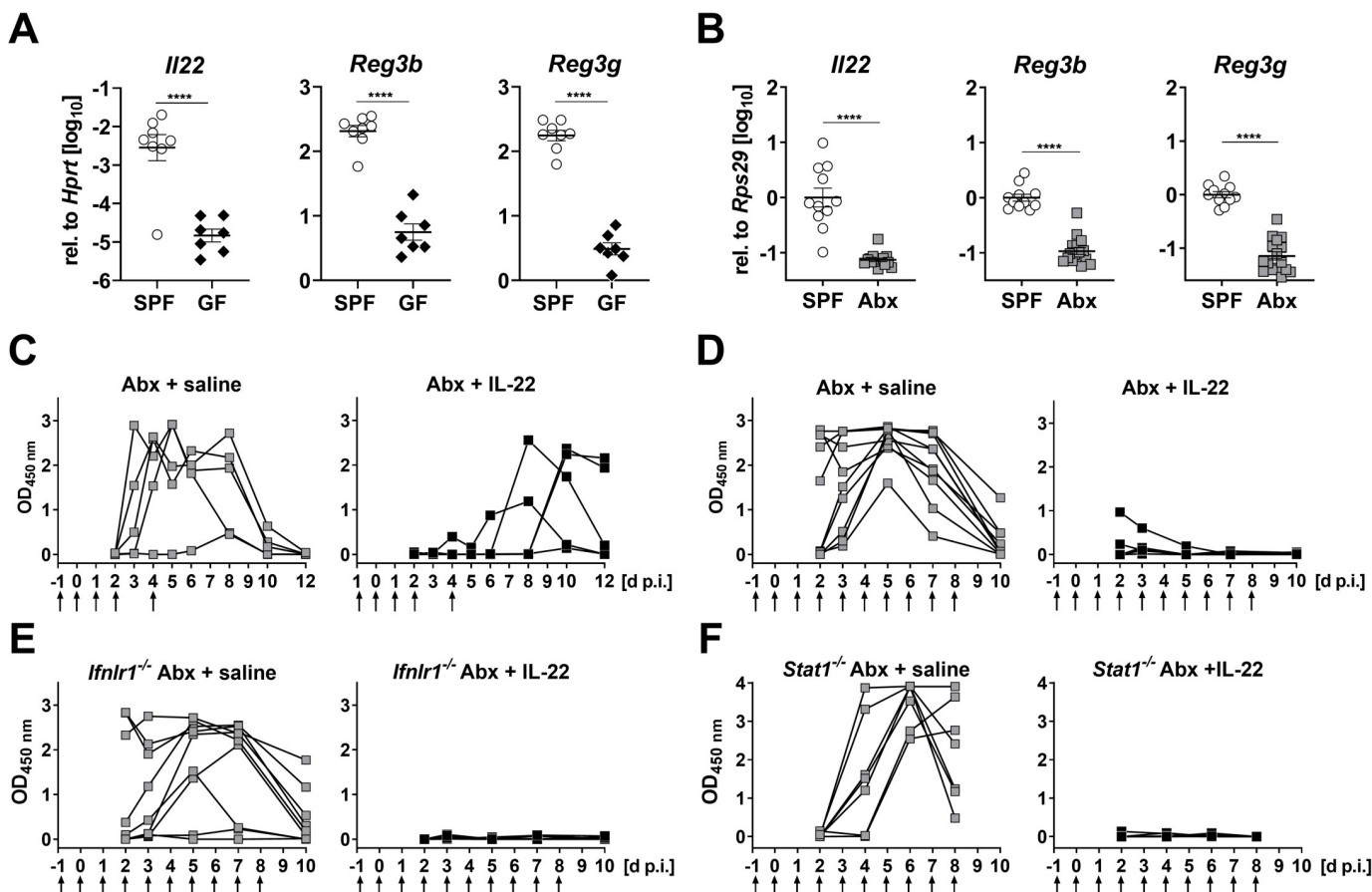

**Fig 3. IL-22 confers short-lived rotavirus resistance in antibiotic-treated C57BL76 mice via a mechanism that is not dependent on IFN signaling.** (A, B) RT-qPCR analysis of tonic expression of *Il22* and IL-22-regulated genes *Reg3b* and *Reg3g* in enriched intestinal epithelial cell fractions of (A) germ-free (GF) or in ileum samples of (B) antibiotic-treated (Abx) mice compared to animals with undisturbed microbiota (SPF). (C, D) Antibiotic-treated WT mice were subjected to brief (n = 5) or extended (n = 10) IL-22 treatment regimens (1 μg per injection, time points indicated by arrows) before infection with $2.4 \times 10^4$ $ID_{50}$ of murine rotavirus strain EDIM by oral gavage. Control animals (C, n = 5; D, n = 9) were treated with saline. Fecal pellets were collected at the indicated time points and viral antigen in the samples was quantified by ELISA. (E, F) Same experimental setup as in panel D, except that (E) *Ifnlr1*$^{-/-}$ (saline n = 9; IL-22 n = 10) or (F) *Stat1*$^{-/-}$ (saline n = 7; IL-22 n = 9) mice were used. Symbols represent individual mice, and bars in (A) and (B) represent means ± SEM. Statistical analysis: Unpaired t-test; ****p<0.0001.

susceptibility of animals with diminished microbiota correlates with diminished baseline expression of IL-22.

We next tested whether IL-22 can confer rotavirus resistance by treating Abx mice with recombinant IL-22 one day before and on days 0, 1, 2 and 4 after oral infection with rotavirus, then monitoring fecal virus shedding by ELISA for the next 12 days. All saline-treated control mice acutely shed high amounts of viral antigen, whereas all IL-22-treated animals exhibited low levels of fecal viral antigen during the first 5 days post-infection (Fig 3C). However, four of the five IL-22-treated mice shed virus with delayed kinetics, indicating that the protective effect of exogenous IL-22 was short-lived. Therefore, in a second experiment, the IL-22 treatment period was extended and daily injections of IL-22 were continued until day 8 post-infection. Under these more stringent conditions, IL-22-mediated suppression of virus shedding was very effective (Fig 3D).

As synergistic activity between IL-22 and IFN-λ to control rotavirus has previously been reported [27], we assessed whether the rotavirus-inhibitory effect of IL-22 was direct or mediated by IFNs. We repeated the IL-22 treatment study using mice in which either IFN-λ

($Ifnlr1^{-/-}$) or all IFN subtypes ($Stat1^{-/-}$) are no longer functional, and found that daily injections of IL-22 were still highly effective at suppressing rotavirus shedding in antibiotic-treated $Ifnlr1^{-/-}$ (Fig 3E) and $Stat1^{-/-}$ (Fig 3F) mice. Thus, IL-22 confers virus resistance in microbiota-depleted mice by a mechanism that is not dependent on IFN-λ or STAT1 signaling.

## Microbes modulate the IL-22 response of the intestinal epithelium

To identify IL-22-induced effector molecules which confer rotavirus resistance in microbiota-depleted mice, we compared gene expression profiles in the ileum of antibiotic-treated $Ifnlr1^{-/-}$ mice which received either saline or IL-22. We identified 22 genes that showed at least 5-fold increased expression and 3 genes with at least 5-fold decreased expression after IL-22 treatment compared with mock-treated mice (Fig 4A). Enhanced expression of several genes,

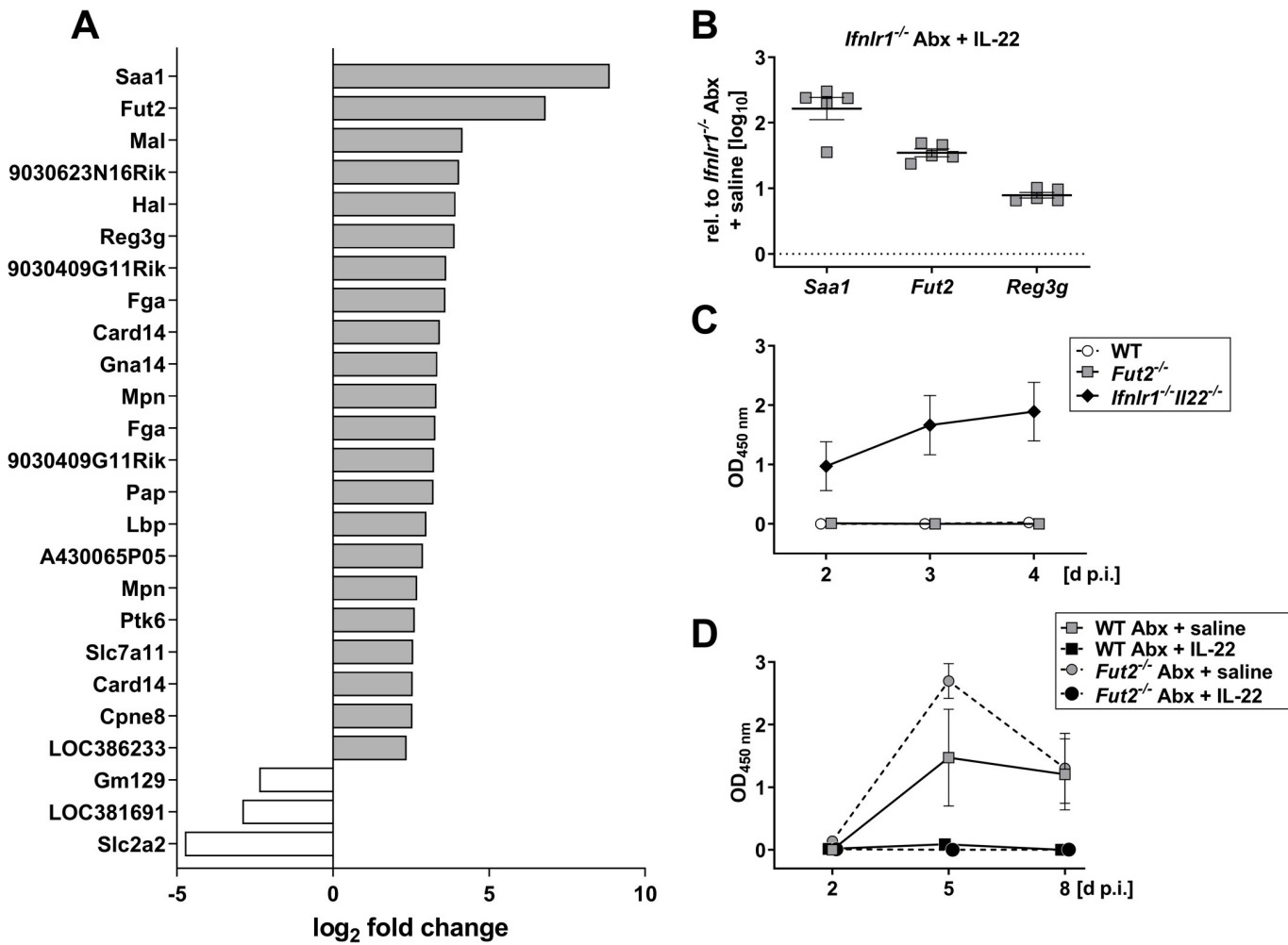

**Fig 4. IL-22 regulates a distinct set of genes in antibiotic-treated $Ifnlr1^{-/-}$ mice, but the $Fut2$ gene plays no decisive role in rotavirus resistance.** (A) Four h prior to preparation of ileum samples for transcriptome analysis, antibiotic-treated $Ifnlr1^{-/-}$ mice (n = 5 per group) were treated with either buffer or 1 μg of IL-22. Genes up- or downregulated at least 5-fold on average in IL-22-treated mice are listed. For all listed genes, variation between samples from the different mice was small (p value <0.01 for every data point) and was part of the original filter used to acquire the data. (B) Verification of IL-22-mediated induction of $Saa1$, $Fut2$ and $Reg3g$ by RT-qPCR. Symbols represent individual mice, and bars represent means ± SEM. (C) $Fut2^{-/-}$ mice with undisturbed microbiota (n = 6) were orally infected with $2.4x10^4$ ID$_{50}$ of murine rotavirus strain EDIM and viral antigen levels in fecal samples were analyzed by ELISA. WT (n = 6) and $Ifnlr1^{-/-}$ $Il22^{-/-}$ (n = 6) mice served as negative and positive controls, respectively. (D) $Fut2^{-/-}$ mice with depleted microbiota (n = 6 per group) were treated daily with saline or IL-22 (extended treatment as described in Fig 3) prior to infection with $2.4x10^4$ ID$_{50}$ of murine rotavirus strain EDIM. Antibiotic-treated WT mice (n = 4–5 per group) served as controls. Symbols in (C) and (D) represent means ± SEM.

including *Saa1*, *Fut2* and *Reg3g*, in tissue of IL-22-treated animals was confirmed by RT-qPCR (Fig 4B).

The *Fut2* gene was of particular interest because it encodes the enzyme fucosyl-transferase 2 (FUT2) that participates in the biosynthesis of histo-blood group antigens which can be recognized by rotavirus [41]. To investigate whether IL-22 inhibits rotavirus growth in the intestine of antibiotic-treated mice by regulating FUT2 expression, we first determined whether *Fut2*$^{-/-}$ mice with normal gut flora were more susceptible to oral infection with rotavirus than standard wild-type mice, using highly susceptible *Ifnlr1*$^{-/-}$*Il22*$^{-/-}$ mice as a positive control (Fig 4C). *Fut2*$^{-/-}$ mice exhibited no defects in rotavirus control. Next, we tested whether IL-22 might no longer be able to inhibit rotavirus replication in antibiotic-treated *Fut2*$^{-/-}$ mice, but found IL-22 treatment to be highly effective in these animals (Fig 4D). We thus concluded that IL-22-mediated upregulation of *Fut2* in microbiota-depleted mice does not play a decisive role in rotavirus resistance.

Since SFB increase the turnover of intestinal epithelial cells that results in rotavirus resistance [20], microbiota-mediated acceleration of infected epithelial cell death might be responsible for the intrinsically high rotavirus resistance of mice with intact microbiota. Therefore, we first evaluated the possibility that the antiviral activity of IL-22 in antibiotic-treated mice results from accelerated death of intestinal epithelial cells by necroptosis [42]. However, we found that IL-22 was effective in inhibiting rotavirus growth in antibiotic-treated *Ripk3*$^{-/-}$ mice (Fig 5A), in which cell death by necroptosis cannot occur [43]. This finding excluded the possibility that IL-22 induces premature death of rotavirus-infected cells by necroptosis in antibiotic-treated mice. An alternative form of induced cell death is pyroptosis, in which caspase-1 plays a key role [44]. To evaluate the possibility that IL-22 acts by inducing pyroptosis in infected cells, possibly via regulation of IL-18 expression [26, 45], we tested IL-22-induced antiviral effects in antibiotic-treated *Casp1/11*$^{-/-}$ mice. We found that IL-22 remained active in these mice (Fig 5B), precluding the possibility that IL-22 inhibits rotavirus by triggering pyroptotic cell death in antibiotic-treated mice.

## Discussion

Our work revealed that the microbiota protects against rotavirus infection and inhibits virus replication in the small intestine. These results are distinct from conclusions of an earlier study [19], which indicated that antibiotic treatment renders mice resistant to rotavirus infection. Of

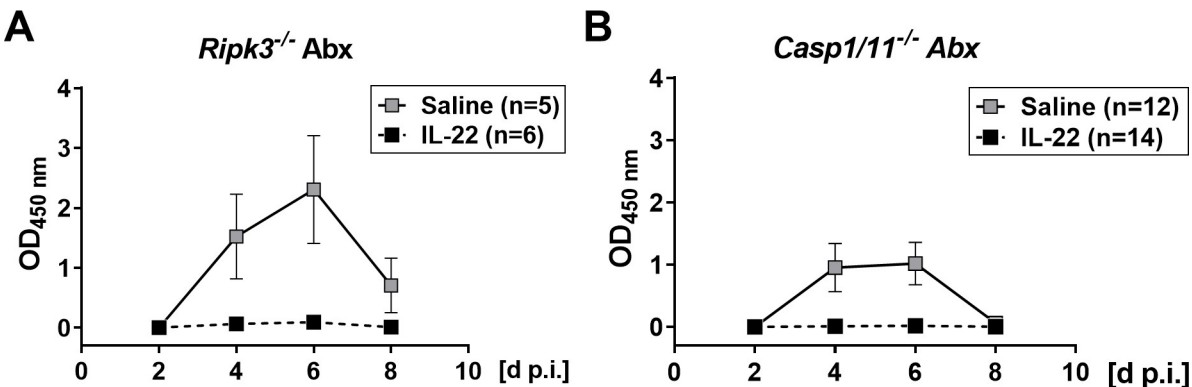

**Fig 5. No role for RIPK3-induced necroptosis or caspase 1-mediated death in IL-22-mediated clearance of rotavirus from antibiotic-treated mice.** Antibiotic-treated *Ripk3*$^{-/-}$ (A) and *Casp1/11*$^{-/-}$ (B) mice were treated with saline or IL-22 and infected with rotavirus strain EDIM as in the experiment described in Fig 3D. Viral antigen levels in fecal samples were analyzed by ELISA. Numbers of animals in each group are indicated. Symbols represent means ± SEM.

interest, our observation of a protective effect of the microbiota against rotavirus recapitulates observations in neonatal gnotobiotic piglets, wherein the microbiota has been shown to protect against diarrhea and viral shedding [46]. Our use of mice from two separate animal facilities located on different continents, as well as two independently-generated rotavirus stocks of EDIM and EC, makes it unlikely that housing conditions or unrecognized mutations in the virus genome are responsible for this discrepancy. We speculate that delayed passage of content through the intestines of antibiotic-treated mice [5, 12, 47] may contribute to differences in virus excretion between microbiota-replete and -depleted mice, which ultimately resulted in differential interpretations regarding the influence of the microbiota on virus resistance. Of note, our results are highly consistent with recent work that attributed the presence of particular SFB with rotavirus resistance of mice [20].

Our study revealed that the rotavirus susceptibility phenotype of antibiotic-treated mice cannot be explained by the involvement of known mediators of rotavirus resistance such as IFN-λ or type I IFN. We confirmed earlier findings [24, 25] that mice with defective IFN receptors are intrinsically more susceptible to rotavirus infection than wild-type mice. Importantly, however, when such mutant mice were treated with antibiotics, their susceptibility to infection with rotavirus still increased, indicating that poorly defined additional factors also contribute to rotavirus resistance. Our analysis suggests that IL-22 plays an important role in this phenomenon. First, we observed that poor expression of IL-22 and IL-22-regulated genes in the ileum correlates with rotavirus susceptibility of antibiotic-treated mice. This observation is consistent with previous work demonstrating decreased intestinal IL-22 expression in the intestine in germ-free or antibiotics-treated mice that can be rescued with short-chain fatty acid administration [48], which drives IL-22 production by CD4+ T cells and innate lymphoid cells [49, 50].

Second, we found that application of IL-22 readily restored rotavirus resistance of antibiotic-treated mice. Interestingly, unlike in mice with undisturbed microbiota in which the antiviral activity of IL-22 was strongly dependent on IFN-λ [27], the anti-rotavirus effect of IL-22 in antibiotic-treated mice was not dependent on IFN-λ or transcription factor STAT1. Thus, IL-22 can negatively affect rotavirus replication by at least two mechanisms, of which only one is dependent on IFN. The IFN-independent mechanism plays a dominant role in antibiotic-treated mice, whereas the IFN-λ-dependent mechanism dominates in mice with undisturbed microbiome.

We currently do not understand how IL-22 inhibits the replication of rotavirus in antibiotic-treated mice. Our transcriptome analysis showed that *Fut2* is strongly induced by IL-22 under such conditions. This finding is in good agreement with earlier work indicating that antibiotic-mediated down-regulation of *Fut2* alters the glycosylation pattern of intestinal epithelial cells [51, 52]. Such changes in protein glycosylation resulted in enhanced susceptibility of mice to infection with several pathogenic bacteria, including *Citrobacter rodentium* [53], Helicobacter pylori [36, 52] and Salmonella typhimurium [51]. However, our experiments with *Fut2*-deficient mice indicate that antiviral defense is not reduced in these animals and that fucosyltransferase-2 is not the elusive IL-22-induced mediator of rotavirus resistance.

IL-22 is known to act through the STAT3 transcription factor in intestinal epithelial cells to drive expression of numerous antimicrobial genes, though it remains unclear which are particularly critical for antiviral effects against rotavirus [26, 54]. Since the intestinal epithelium consists of short-lived cells, we envisaged the possibility that IL-22 might inhibit rotavirus replication by inducing premature death of infected cells. Yet, we found that IL-22 effectively blocked rotavirus replication in antibiotic-treated mice lacking key factors for necroptosis or pyroptosis, rendering such a mechanism unlikely. However, since these programmed cell death pathways are highly interconnected, we cannot exclude the possibility of compensatory

effects [55]. Available data suggest that SFB prevents and cures rotavirus infection by enhancing the proliferation of infected epithelial cells in the terminal ileum, promoting their shedding from the tissue [20]. We speculate that IL-22 may provide similar growth-promoting stimuli to epithelial cells in antibiotic-treated mice and that enhanced cell proliferation might explain the antiviral effect of IL-22 that we observed. It has also been suggested that IL-22 may induce extrusion of intestinal epithelial cells [30], which remains a possibility in the context of microbiota-mediated regulation of rotavirus infection. Recently, reversal of viral appropriation of autophagic flux has been suggested as an antiviral mechanism for IL-22 in control of respiratory syncytial virus, a possibility that remains open for its regulation of rotavirus [56]. Our study reveals a clear role for the microbiota in maintenance of a homeostatic antiviral state, driven by IL-22, against rotavirus infection, and future studies delineating the specific pathways required for this activity may help reveal additional therapeutic approaches against this pathogen.

## Acknowledgments

We thank the Clean Mouse Facility (CMF), University of Bern, Bern, Switzerland, for providing GF mice, Ana Magalhães, University of Porto, Portugal for providing *Fut2*-deficient mice, and Otto Haller for helpful comments on the manuscript.

## Author Contributions

**Conceptualization:** Laure Dumoutier, Andreas Diefenbach, Andreas Wack, Megan T. Baldridge, Peter Staeheli.

**Data curation:** Daniel Schnepf, Tanel Mahlakõiv, Stefania Crotta, Andreas Wack.

**Formal analysis:** Daniel Schnepf, Pedro Hernandez, Tanel Mahlakõiv, Andreas Wack.

**Funding acquisition:** Megan T. Baldridge, Peter Staeheli.

**Investigation:** Daniel Schnepf, Pedro Hernandez, Tanel Mahlakõiv, Stefania Crotta, Meagan E. Sullender, Stefan T. Peterson, Annette Ohnemus.

**Methodology:** Daniel Schnepf, Pedro Hernandez, Tanel Mahlakõiv, Meagan E. Sullender, Stefan T. Peterson, Annette Ohnemus.

**Resources:** Camille Michiels, Ian Gentle, Laure Dumoutier, Celso A. Reis.

**Supervision:** Andreas Diefenbach, Megan T. Baldridge, Peter Staeheli.

**Validation:** Daniel Schnepf.

**Visualization:** Daniel Schnepf, Stefania Crotta.

**Writing – original draft:** Megan T. Baldridge, Peter Staeheli.

**Writing – review & editing:** Andreas Wack, Peter Staeheli.

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
