## [Decision Letter · Decision Letter 0]

18 Mar 2021

PONE-D-21-04729

Rotavirus susceptibility of antibiotic-treated mice ascribed to diminished expression of interleukin-22

PLOS ONE

Dear Dr. Staeheli,

Thank you for submitting your manuscript to PLOS ONE. After careful consideration, we feel that it has merit but does not fully meet PLOS ONE’s publication criteria as it currently stands. Therefore, we invite you to submit a revised version of the manuscript that addresses the points raised by three experts who reviewed your study.

We look forward to receiving your revised manuscript.

Kind regards,

Michael Nevels

Academic Editor

PLOS ONE

Journal Requirements:

Reviewers' comments:

Reviewer's Responses to Questions

**Comments to the Author**

1. Is the manuscript technically sound, and do the data support the conclusions?

Reviewer #1: No

Reviewer #2: Yes

Reviewer #3: Partly

2. Has the statistical analysis been performed appropriately and rigorously? 

Reviewer #1: Yes

Reviewer #2: Yes

Reviewer #3: No

3. Have the authors made all data underlying the findings in their manuscript fully available?

Reviewer #1: Yes

Reviewer #2: Yes

Reviewer #3: Yes

4. Is the manuscript presented in an intelligible fashion and written in standard English?

Reviewer #1: Yes

Reviewer #2: Yes

Reviewer #3: Yes

5. Review Comments to the Author

Reviewer #1: This work presents data that indicates that C57BL/6 mice are highly, if not completely resistant to infection by WT murine rotavirus and that this resistance can be alleviated by ablation of microbiota by either antibiotics or germfree approaches. They then go on to show that such treatments reduce IL-22 and that exogenous IL-22 protects against rotavirus, and that all of these findings are independent of type 1 IFN. There is not really all that much new here but, nonetheless, the work cold revised to make a reasonable contribution to the literature but major revisions in how they present and interpret their findings are needed.

Infection of C57BL/6 mice with murine rotavirus EC strain is a very widely used model of rotavirus infection, used for many years. Accordingly, when such mice are directly purchased from most commercial vendors and bred at many institutions, they will display clear shedding of RV antigens and genomes. Here, the authors report that such mice bred at 2 different institutions were not infectable by RV unless first exposed to antibiotics. I don’t doubt the observation they made but how they present it and interpret it is highly problematic. Firstly, they don’t actually acknowledge that the untreated mice were not, in fact, infected with RV at all and thus that they their positive control did not work as it has in the literature for over 20 years. Consequently, they have not really interpreted and properly contextualized the antibiotic data. The proper interpretation is that, for some reason, their mice are highly RV resistant, relative to mice long used in the literature which can be readily infected by this RV strain. They mention “timing of virus excretion” as a possible explanation but I have no idea what they mean by this. Countless studies have shown adult C57BL/6 shed mice within a few days of inoculation and their assay period certainly covers this time. It is certainly appropriate to cite reference 19 to speculate that the mice studied here might also be RV-resistant to their specific microbiota composition but again they need to start by acknowledging that their mice contradict the well-established phenotype. Moreover, this makes it very difficult to compare their findings with published papers that started with the typical readily-infectable mice and then examined impacts of antibiotics and germfree conditions.

Reviewer #2: Dear Authors,

It is a very important scientific work and would definitely contribute in the advancement of the current knowledge about the influence of microbiome and its corelation with antibiotics on viral infections.

I would strongly suggest you the improvement of all figures. I am unable to clearly read what is written along x axis and y axis. Kindly improve it. It is in bold letters, it will be more convenient for the readers if it appears in normal letters.

Moreover at line number 142 (Antibiotic treatment of Mice), kindly mention the mode of adminstration of antibiotics, as I could not get it completely. you are requested to make it clear weather the antibiotics were adminsitered orally or intramuscular or intravenous.

The rest is fine to my understanding.

Reviewer #3: The manuscript titled “Rotavirus susceptibility of antibiotic-treated mice ascribed to diminished expression of IL-22” by Schnepf et al describes the finding that rotavirus, a common intestinal viral infection, replicates more efficiently in the intestines of germ free and antibiotic treated mice compared to wild type mice. The authors ascribe this finding to the fact that the microbiome is altered in the germ free and antibiotic treated mice. The effect was associated with the diminished expression of IL-22 and when IL-22 was administered to germ free mice, the replication of the virus was blocked presenting with a more wild type phenotype. They present several experiments that exclude specific pathways that could explain this finding including interferon, glycosylation patterns, necroptosis, and pyroptosis. Although the findings are interesting and provocative, the authors do not provide an explanation for their findings beyond the association with IL22. There is a vast volume of literature in the murine model of rotavirus over the last 35 years and a recent publication from Shi et al (Cell 2019) that directly contradict what is presented here. This descriptive observation, which lacks full development and explanation, adds little to the scientific field. In addition, discrepancies in the experimental approach raises some concerns.

Major Concerns:

This paper needs several crucial modifications before it can be published. Overall, the introduction needs to be more informative about rotavirus-specific interactions with bacteria and IL-22. The presented results are strong and the discovery that IL-22 is dependent on gut microbiota is excellent but more information is needed about the strain of rotavirus used for each experiment and more discussion is needed about variation in susceptibility of replicate mice. The discussion largely repeats the results and should instead be put into context with the other viral interactions currently in the discussion. Although the mechanism remains elusive, the authors do not postulate a possible pathway based on literature detailing what is known about the interplay between bacteria and immune effectors.

1. This paper directly contradicts over 35 years of work in the murine model by at least 5 major research groups and many more smaller groups. Recently, it was published that antibiotic treatment reduced rather than promoted rotavirus infection (Shi et al Cell 2019). Although the authors do acknowledge this publication, a more depth discussion is necessary to provide explanation beyond IL22 as to why these data differ from the vast volume of previous studies.

2. The gnotobiotic piglet models (Yuan et all JV 1998) has been used for many years and does not seem to model the authors findings here. Discussing the findings in the context of these published results is warranted and should be included in the discussion.

3. Lines 238-286: Although the authors are looking at the immune effectors that contribute to this resistance rather than specific microbiome composition, and they reference Shi et all Cell 2019, the authors do not test for the presence of SFB. Thus it is not clear whether this work is extending the findings of Shi et al or the authors have found a completely independent protective mechanism from RV. The mice should be tested for the presence of SFB either through SFB-specific qPCR or another 16S sequencing approach. More evidence that this extends the work of Shi et al is needed before publication. Rewording the introduction and the discussion is recommended as well as adding qPCR specific for SFB.

4. There are major differences in virulence between the EDIM strain and the EC strain. An explanation should be included in the discussion as to the differences in virulence of these strains and how it might affect the interpretation of the results. Line 239 and 248: Since authors are using 2 different strains of rotavirus, they should specify which strain they are referring to (EDIM or EC). Also reference strain in the figure legends

5. Microarray is an outdated method in which to assess transcriptional changes. State of the art approaches should utilize RNAseq based transcriptomics

6. Since viral stocks were made from intestinal and fecal isolates that were rich in microbial organisms, some description is necessary as to how the authors ensured that the inoculum was bacteria free.

7. Immunofluorescence for RV antigen in intestinal sections should be included to demonstrate there is no infection in the wild type animals.

Minor Concerns:

1. With antibiotics in the drinking water, how was the amount of antibiotic that each animal received controlled for and how would this affect the microbiome differently in each animal. This could perhaps explain the large differences in the shedding curves of each animal.

2. 16S sequencing should be done before and after antibiotic treatment in order to verify that large changes in the microbiome were occurring with treatment. The different mouse strains used may not have comparable microbiomes

3. Rotavirus is a small intestinal infection and so it is not clear why colonic samples were used for the viral stock of EDIM rather than small intestine. An explanation should be included in the materials and methods.

4. An upaired student’s t test does not seem the appropriate test to assess statistical significance. Studies using this statistical test should have more than six samples in each group and with this data, the ability to generalize to a larger population is difficult. Appropriate statistical tests should be performed.

5. It is not clear whether all the animals were on the same food and water. This would be important as it could explain some of the differences the authors observed. The description of food and water should be included in the materials and methods.

6. Were the animals prechecked for the presence of serum RV antibodies that might suggest the animals had previously seen RV? This should be done and stated in the materials and methods.

7. Line 291 Why were IFNLR1-/- mice used instead of WT mice for these studies. More rationale should be included.

8. In Figure 3 A and B why is there such a large difference in expression levels in the SPF controls. A discussion of this should be included in the results.

9. Figures 3 C and D need complete shedding curves and area under the curve calculated as performed in O’Neal et all Virology 1997.

10. In Figure 2, day 5 seems too late to assess viral shedding based on the work of Offit, Greenberg, Ward, Estes, Connor. Earlier timepoints should be presented.

11. What does the statement in lines 325-326 mean about the timing? The timing of shedding of rotavirus in the murine model of rotavirus infection has been well documented over the last 30 years by multiple groups (Offit, Greenberg, Ward, Estes, Connor)

12. Line 94-98: The authors should mention that IL-22 is known to be protective from rotavirus infection (reference 18)

13. Line 242 and 249: Loss of infectivity in C57BL6 adult mice is highly surprising. The SFB protection from RV reported by Shi et al was the first identification of a resistant mouse model. A short statement such as “this was surprising given reports of consistent infection in previous studies” should be added to put the discovery in context for readers.

14. Line 249 and Figure 2: The authors should make it clear whether the baseline of 3e10 genome copies per fecal pellet is the limit of detection or the expected amount of RV shedding. It is unclear whether the mice at the second independent institution were also resistant to RV infection or simply had an increase from baseline infectivity.

15. Line 255-256 and Figure 2: No reference is made to the variability in the -Abx condition, despite the fact that this hints at variation either in the microbiota composition or the IL-22 signaling. Some discussion on this point should be included. Comparison between mice that are highly susceptible and those that are less susceptible may even be helpful in determining the mechanism.

16. Line 265-266: It should be clarified that these are basal levels in the absence of infection

17. Line 295-304: As Fut2 enables RV entry, it is not clear why an increase in Fut2 would confer resistance to RV infection. More rationale should be included to understand the logic behind this statement.

18. Line 315 and Figure 5B: The authors do not discuss why there is reduced shedding in saline treated Casp1/11-/- Abx mice as compared to other mouse models treated with Abx. A discussion should be included to address this.

19. Figure 5 legend: The question mark following pyroptosis is unnecessary.

20. Lines 46 and 296: The comments about O glycosylation are incorrect. Fut 2 adds fucose moieties to glycoproteins and glycolipid so they are fucosylated glycans on the cell surface. This enzyme has nothing to do with O glycans. There are no O molecules linked to rotavirus entry. The wording should be modified

6. PLOS authors have the option to publish the peer review history of their article (what does this mean?). If published, this will include your full peer review and any attached files.

Reviewer #1: No

Reviewer #2: No

Reviewer #3: No

---

## [Author Response · Author response to Decision Letter 0]

28 Apr 2021

We wished to thanks the reviewers for their constructive criticism. We addressed all concerns as detailed in the rebuttal letter. The various text changes clearly improved the quality of our manuscript and will facilitate assessing the main scientific conclusions resulting from our work.

Rebuttal letter:

Reviewer #1: 

This work presents data that indicates that C57BL/6 mice are highly, if not completely resistant to infection by WT murine rotavirus and that this resistance can be alleviated by ablation of microbiota by either antibiotics or germfree approaches. They then go on to show that such treatments reduce IL-22 and that exogenous IL-22 protects against rotavirus, and that all of these findings are independent of type 1 IFN. There is not really all that much new here but, nonetheless, the work cold revised to make a reasonable contribution to the literature but major revisions in how they present and interpret their findings are needed.

Infection of C57BL/6 mice with murine rotavirus EC strain is a very widely used model of rotavirus infection, used for many years. Accordingly, when such mice are directly purchased from most commercial vendors and bred at many institutions, they will display clear shedding of RV antigens and genomes. Here, the authors report that such mice bred at 2 different institutions were not infectable by RV unless first exposed to antibiotics. I don’t doubt the observation they made but how they present it and interpret it is highly problematic. Firstly, they don’t actually acknowledge that the untreated mice were not, in fact, infected with RV at all and thus that they their positive control did not work as it has in the literature for over 20 years. Consequently, they have not really interpreted and properly contextualized the antibiotic data. The proper interpretation is that, for some reason, their mice are highly RV resistant, relative to mice long used in the literature which can be readily infected by this RV strain. 

Response: We would like to clarify that we did not intend to claim that our WT mice are “not infectable” by RV unless treated with antibiotics. It has previously been reported by multiple other groups that adult C57BL/6 mice infected with EDIM or EC exhibit “low and inconsistent levels of” viral shedding (PMID 15003858), and that “the ID50 for C57BL/6 mice was approximately 1000× the dose of ECwt required for the BALB/c mice” (PMID 16191453), consistent with our observations of low viral shedding in adult C57BL/6 mice at two independent sites. The key aspect of these experiments we intended to emphasize that our data demonstrate that RV replicates much better in germ-free or antibiotic-treated mice than in mice with an undisturbed microbiota. We apologize for not having emphasized this important point well enough. We rephrased the corresponding sections of the revised manuscript (current lines 270-274). 

They mention “timing of virus excretion” as a possible explanation but I have no idea what they mean by this. 

Response: We suggest that differences in “timing of virus excretion” between microbiota-replete and -depleted mice might explain the discrepancy between our data and published work by other authors. It has been reported that antibiotic treatment can delay the passage of content through the intestinal tract of mice (PMID 25431490, 21998395, 24237703). In the revised manuscript (lines 342-345), we rephrased the critical sentence to clarify this point, and cite the work which demonstrated reduced speed of content passage in mice with depleted microbiota. 

Countless studies have shown adult C57BL/6 shed mice within a few days of inoculation and their assay period certainly covers this time. It is certainly appropriate to cite reference 19 to speculate that the mice studied here might also be RV-resistant to their specific microbiota composition but again they need to start by acknowledging that their mice contradict the well-established phenotype. Moreover, this makes it very difficult to compare their findings with published papers that started with the typical readily-infectable mice and then examined impacts of antibiotics and germfree conditions.

Response: As discussed above, multiple other groups have also reported low levels of viral shedding in adult C57BL/6 mice, and we have now clarified that our WT mice “shed only low amounts of virus” unless treated with antibiotics. We have also added in the statement, “Since our animal facilities are not free of segmented filamentous bacteria (SFB) which are known to confer rotavirus resistance [20], we speculate that the degree of SFB colonization in individual animals could regulate intrinsic rotavirus susceptibility” (lines 271-274) to emphasize the possibility of already-present microbial factors that could contribute to low levels of viral infection in mice with replete microbiota.

Reviewer #2: 

It is a very important scientific work and would definitely contribute in the advancement of the current knowledge about the influence of microbiome and its corelation with antibiotics on viral infections.

I would strongly suggest you the improvement of all figures. I am unable to clearly read what is written along x axis and y axis. Kindly improve it. It is in bold letters, it will be more convenient for the readers if it appears in normal letters.

Response: High-resolution versions of the figures were provided, but required downloading from the PLoS One server. These versions contain axis labels that are easily readable.

Moreover at line number 142 (Antibiotic treatment of Mice), kindly mention the mode of adminstration of antibiotics, as I could not get it completely. You are requested to make it clear weather the antibiotics were adminsitered orally or intramuscular or intravenous.

Response: We apologize for any confusion. Antibiotics were added to the drinking water at concentrations described in the methods section. We rephrased this section to now make the administration method clearer (lines 151-158).

Reviewer #3: 

The manuscript titled “Rotavirus susceptibility of antibiotic-treated mice ascribed to diminished expression of IL-22” by Schnepf et al describes the finding that rotavirus, a common intestinal viral infection, replicates more efficiently in the intestines of germ free and antibiotic treated mice compared to wild type mice. The authors ascribe this finding to the fact that the microbiome is altered in the germ free and antibiotic treated mice. The effect was associated with the diminished expression of IL-22 and when IL-22 was administered to germ free mice, the replication of the virus was blocked presenting with a more wild type phenotype. They present several experiments that exclude specific pathways that could explain this finding including interferon, glycosylation patterns, necroptosis, and pyroptosis. Although the findings are interesting and provocative, the authors do not provide an explanation for their findings beyond the association with IL22. 

Response: The last notion is certainly correct: we did not manage to elucidate the mechanism by which IL-22 limits rotavirus replication in the intestinal tract of antibiotic-treated mice. However, we addressed several hypotheses which, unfortunately, proved invalid. A better understanding of this interesting new activity of IL-22 at the molecular level can hopefully be achieved in future studies.

There is a vast volume of literature in the murine model of rotavirus over the last 35 years and a recent publication from Shi et al (Cell 2019) that directly contradict what is presented here. 

Response: We respectfully disagree with the view that our work would directly contradict the findings of Shi and coworkers. These researchers reported that specific segmented filamentous bacteria (SFB) that heavily colonizes the intestine of certain mouse strains are able to confer a very high degree of RV resistance to immunocompromised (Rag1-/-) mice, a resistance which is not mediated by IL-22. However, these authors did not dispute earlier findings of several other labs that IL-22 can impede RV replication in the intestinal tract of mice, nor did this work address the cumulative effects of the endogenous microbiota in regulating RV infection. 

This descriptive observation, which lacks full development and explanation, adds little to the scientific field. In addition, discrepancies in the experimental approach raises some concerns.

Response: We are not sure which discrepancies are meant. 

Major Concerns:

This paper needs several crucial modifications before it can be published. Overall, the introduction needs to be more informative about rotavirus-specific interactions with bacteria and IL-22. 

Response: The introduction has been substantially expanded to now include additional details and references related to bacteria-rotavirus interactions, as well as to further discuss what is known to date about IL-22-mediated antiviral effects upon rotavirus. 

The presented results are strong and the discovery that IL-22 is dependent on gut microbiota is excellent but more information is needed about the strain of rotavirus used for each experiment and more discussion is needed about variation in susceptibility of replicate mice.

Response: Detailed information on the rotavirus strains used in each particular figure is now provided in the figure legends. We have also now added statements addressing the variation among mice with the following: “Rotavirus shedding by untreated mutant mice was surprisingly heterogeneous (Fig. 2), raising the possibility that the microbiota composition of individual animals could matter. Since our animal facilities are not free of segmented filamentous bacteria (SFB) which are known to confer rotavirus resistance [20], we speculate that the degree of SFB colonization in individual animals could regulate intrinsic rotavirus susceptibility.”

The discussion largely repeats the results and should instead be put into context with the other viral interactions currently in the discussion. Although the mechanism remains elusive, the authors do not postulate a possible pathway based on literature detailing what is known about the interplay between bacteria and immune effectors.

Response: The discussion has been expanded to include additional discussion of the mechanisms by which the microbiota maintains IL-22 expression.

1. This paper directly contradicts over 35 years of work in the murine model by at least 5 major research groups and many more smaller groups. Recently, it was published that antibiotic treatment reduced rather than promoted rotavirus infection (Shi et al Cell 2019). Although the authors do acknowledge this publication, a more depth discussion is necessary to provide explanation beyond IL22 as to why these data differ from the vast volume of previous studies. 

Response: These authors showed that GSU-RAG mice (which carry SFB that block infection with rotavirus) could not be rendered rotavirus susceptible by prolonged treatment with antibiotics. However, as mentioned above, our wild-type mice are not fully resistant to rotavirus infection. SFB is present in most mice of our colony, and recent PCR data indicate that the extent of SFB colonization is most prominent in immune-deficient mice. We now discuss this finding in the revised manuscript (lines 270-274). Nevertheless, we continue to believe that the limited RV susceptibility of our mice cannot be explained entirely by the presence of SFB in our colonies mainly because, unlike Shi and coworkers, we do not observe complete resistance to RV infection. However, the unexplained heterogeneity of RV susceptibility of some individual mice in our experiments could well be due to a higher abundance of SFB in these individuals. To account for these observations, we rephrased the text to indicate that the role of SFB remains unclear in our experimental setting.

2. The gnotobiotic piglet models (Yuan et all JV 1998) has been used for many years and does not seem to model the authors findings here. Discussing the findings in the context of these published results is warranted and should be included in the discussion.

Response: We would suggest that findings from gnotobiotic piglets in fact nicely recapitulate what we have observed in mice. Specifically, it has been reported that colonized piglets exhibit reduced HRV-induced diarrhea and viral shedding compared to their noncolonized germ-free counterparts, which indicates a protective role for the microbiota against RV (PMID 29929472). This is highly compatible with our observations, and we agree that this helpful point should be added to the discussion (lines 336-339).

3. Lines 238-286: Although the authors are looking at the immune effectors that contribute to this resistance rather than specific microbiome composition, and they reference Shi et all Cell 2019, the authors do not test for the presence of SFB. Thus it is not clear whether this work is extending the findings of Shi et al or the authors have found a completely independent protective mechanism from RV. The mice should be tested for the presence of SFB either through SFB-specific qPCR or another 16S sequencing approach. More evidence that this extends the work of Shi et al is needed before publication. Rewording the introduction and the discussion is recommended as well as adding qPCR specific for SFB.

Response: As proposed by this reviewer, we tested for the presence of SFB in our mice and found that the majority of animals in our two facilities are colonized by these bacteria. These results are mentioned in the revised manuscript and possible implications of these findings are discussed (lines 270-274).

4. There are major differences in virulence between the EDIM strain and the EC strain. An explanation should be included in the discussion as to the differences in virulence of these strains and how it might affect the interpretation of the results. Line 239 and 248: Since authors are using 2 different strains of rotavirus, they should specify which strain they are referring to (EDIM or EC). Also reference strain in the figure legends

Response: We have now more clearly specified which virus strains were used for the experiment in the various graphs, and have included the point that we observed similar findings for both EDIM and EC in our discussion.

5. Microarray is an outdated method in which to assess transcriptional changes. State of the art approaches should utilize RNAseq based transcriptomics

Response: We agree that microarrays represent an outdated method. However, at the time when this particular experiment was performed, this technique was still used in many laboratories. Since this approach yielded clear hits which could be validated by standard RT-qPCR technology, the resulting information was useful and triggered the evaluation of new hypotheses. Although state of the art approaches might be more sensitive and might yield additional hits, we do not believe that such alternative technology could provide entirely novel insights with regard to IL-22-mediated gene regulation in our system. 

6. Since viral stocks were made from intestinal and fecal isolates that were rich in microbial organisms, some description is necessary as to how the authors ensured that the inoculum was bacteria free.

Response: The methods section contains this information: the virus stocks were filtered before use.

7. Immunofluorescence for RV antigen in intestinal sections should be included to demonstrate there is no infection in the wild type animals.

Response: As discussed above, we have now fully clarified in the text that our WT mice with intact microbiota are not fully resistant to infection with RV, and are instead focused in this study on the increased viral levels observed when the microbiota is depleted. 

Minor Concerns:

1. With antibiotics in the drinking water, how was the amount of antibiotic that each animal received controlled for and how would this affect the microbiome differently in each animal. This could perhaps explain the large differences in the shedding curves of each animal.

Response: We did not monitor the drinking behavior of individual animals. However, since the antibiotic cocktail was the only source of water and since the antibiotic treatment lasted several weeks, we assume that sufficient amounts of antibiotics were present in all mice. Prior experiments using the same methods have demonstrated consistent depletion of the microbiota below the limit of detection of assays (PMID 25431490). In most experiments we confirmed the virtual absence of live bacteria in the feces by plating the material onto suitable agar petri dishes. This latter fact is mentioned in the revised manuscript (lines 157-158).

2. 16S sequencing should be done before and after antibiotic treatment in order to verify that large changes in the microbiome were occurring with treatment. The different mouse strains used may not have comparable microbiomes

Response: As mentioned above, in most experiments we confirmed the efficacy of treatment by plating fecal samples on BHI agar-containing petri dishes, and the capacity of antibiotics treatment to dramatically deplete WT and Ifnlr1-/- mice of bacteria has been previously demonstrated (PMID 25431490). While microbiota compositions are broadly similar between WT and Ifnlr1-/- mice (PMID 25431490), we have not carefully compared the initial microbiota composition of some mouse strains used here in our facilities. Although we cannot exclude variations between strains, it is unlikely that such differences in the microbiota could affect our principal conclusion that microbiota-driven production of IL-22 can limit RV replication in the intestinal tract.

3. Rotavirus is a small intestinal infection and so it is not clear why colonic samples were used for the viral stock of EDIM rather than small intestine. An explanation should be included in the materials and methods.

Response: Although RV predominately replicates in the small intestine, large amounts of virus are present in the stool of infected pups. Therefore, to produce virus stocks, we used the complete intestine including both small intestine and colon with content as virus source, a method described clearly in the materials and methods section. This approach is consistent with a general preference by our institutions for avoiding use of excessive mice.

4. An upaired student’s t test does not seem the appropriate test to assess statistical significance. Studies using this statistical test should have more than six samples in each group and with this data, the ability to generalize to a larger population is difficult. Appropriate statistical tests should be performed.

Response: Here, statistical analyses with unpaired student’s t test were performed exclusively with data shown in figures 3A and 3B. In these experiments, the group sizes were larger than 6, consistent with the reviewer’s point.

5. It is not clear whether all the animals were on the same food and water. This would be important as it could explain some of the differences the authors observed. The description of food and water should be included in the materials and methods.

Response: The same diet was used for all animals, but some animals received antibiotics with the drinking water. This fact has now been stated more clearly in the revised manuscript (lines 151-158). 

6. Were the animals prechecked for the presence of serum RV antibodies that might suggest the animals had previously seen RV? This should be done and stated in the materials and methods.

Response: No testing for the presence of RV-specific antibodies was performed. Animals were housed in a clean facility that is routinely screened for the presence of RV using sentinel mice with no evidence for spontaneous RV infections. This fact is mentioned in the revised manuscript (lines 138-141). 

7. Line 291 Why were IFNLR1-/- mice used instead of WT mice for these studies. More rationale should be included.

Response: We used Ifnlr1-ko mice for this experiment to exclude possible complications from the fact that IL-22 can act by influencing the activity of IFN-λ (ref. 27). We have detailed this rationale in our manuscript text. 

8. In Figure 3 A and B why is there such a large difference in expression levels in the SPF controls. A discussion of this should be included in the results.

Response: Data shown in Figure 3A was derived from analyzing enriched intestinal epithelial cell fractions and results were normalized to the housekeeping gene Hprt. Data shown in Figure 3B was derived from ileum samples and results were normalized to a different housekeeping gene (Rps29). Therefore, absolute Il22 expression levels cannot be assessed by simply comparing Figures 3A and B.

9. Figures 3 C and D need complete shedding curves and area under the curve calculated as performed in O’Neal et all Virology 1997.

Response: The shedding curves cover the complete observation period of 12 days. We believe that the graph nicely illustrates our findings and that no additional calculations are required.

10. In Figure 2, day 5 seems too late to assess viral shedding based on the work of Offit, Greenberg, Ward, Estes, Connor. Earlier timepoints should be presented.

Response: In our hands, day 5 post-infection proved to be an effective time point for assessing differences in fecal RV shedding. 

11. What does the statement in lines 325-326 mean about the timing? The timing of shedding of rotavirus in the murine model of rotavirus infection has been well documented over the last 30 years by multiple groups (Offit, Greenberg, Ward, Estes, Connor)

Response: We suggest that differences in “timing of virus excretion” between microbiota-replete and -depleted mice might explain the discrepancy between our data and published work by other authors. It has been reported that antibiotic treatment can delay the passage of content through the intestinal tract of mice (PMID 25431490, 21998395, 24237703). In the revised manuscript (lines 342-345), we rephrased the critical sentence to clarify this point, and cite the work which demonstrated reduced speed of content passage in mice with depleted microbiota.

12. Line 94-98: The authors should mention that IL-22 is known to be protective from rotavirus infection (reference 18)

Response: In the introduction we have further expanded our discussion of the relevant literature which indicates that IL-22 can protect from RV infection. 

13. Line 242 and 249: Loss of infectivity in C57BL6 adult mice is highly surprising. The SFB protection from RV reported by Shi et al was the first identification of a resistant mouse model. A short statement such as “this was surprising given reports of consistent infection in previous studies” should be added to put the discovery in context for readers.

Response: We did not report a “loss of infectivity” in C57BL/6 mice. We demonstrate that if WT mice with standard microbiota were orally infected with a given dose of RV, we could detect how levels of virus in fecal samples by ELISA. However, virus shedding was detected in the majority of WT mice if they were treated with antibiotics or if germ-free animals were used for the experiments. Based upon previous reports demonstrating poor infection of C57BL/6 mice by other groups, discussed above, we were not particularly surprised to find low levels of viral infection in our WT mice with replete microbiota.

14. Line 249 and Figure 2: The authors should make it clear whether the baseline of 3e10 genome copies per fecal pellet is the limit of detection or the expected amount of RV shedding. It is unclear whether the mice at the second independent institution were also resistant to RV infection or simply had an increase from baseline infectivity.

Response: As mentioned above, we did not claim that our mice are completely resistant to infection with RV if they had a normal microflora. We used experimental conditions under which mice with standard microbiota showed a very low level of viral infection. Under these same conditions, mice with diminished microbiota were highly susceptible to RV infection. We have now further clarified that the line represents the limit of detection of the RT-qPCR assay in the figure legend. 

15. Line 255-256 and Figure 2: No reference is made to the variability in the -Abx condition, despite the fact that this hints at variation either in the microbiota composition or the IL-22 signaling. Some discussion on this point should be included. Comparison between mice that are highly susceptible and those that are less susceptible may even be helpful in determining the mechanism.

Response: The Freiburg and St. Louis laboratories used slightly different antibiotics cocktails to effectively suppress the gut microbiota. In both laboratories we noted some heterogeneity with few individual animals in almost every group showing different rotavirus susceptibility. We do not understand the molecular basis of this phenomenon and we did not try to link it to possible changes in the composition of the microbiota of individual mice. We agree further studies examining the source of heterogeneity among similar mice would be of interest, but is beyond the scope of our current study.

16. Line 265-266: It should be clarified that these are basal levels in the absence of infection

Response: As suggested by this reviewer, we modified the text to better indicate that non-infected mice were analyzed (lines 281-282). 

17. Line 295-304: As Fut2 enables RV entry, it is not clear why an increase in Fut2 would confer resistance to RV infection. More rationale should be included to understand the logic behind this statement.

Response: The working hypothesis for this particular experiment was rephrased to better explain the rationale (lines 311-312).

18. Line 315 and Figure 5B: The authors do not discuss why there is reduced shedding in saline treated Casp1/11-/- Abx mice as compared to other mouse models treated with Abx. A discussion should be included to address this.

Response: Since no side-by-side experiments were performed, we do not know whether virus shedding of antibiotic-treated Casp1/11-ko mice was reduced compared with antibiotic-treated WT mice. This particular experiment did not aim at answering this particular question. Rather, this experiment aimed at answering the question whether IL-22 remained active in antibiotic-treated Casp1/11-ko mice. Our results clearly demonstrate that this was the case. 

19. Figure 5 legend: The question mark following pyroptosis is unnecessary.

Response: Thanks for noting this mistake, which has been corrected in the revised manuscript.

20. Lines 46 and 296: The comments about O glycosylation are incorrect. Fut 2 adds fucose moieties to glycoproteins and glycolipid so they are fucosylated glycans on the cell surface. This enzyme has nothing to do with O glycans. There are no O molecules linked to rotavirus entry. The wording should be modified

Response: We rephrased the critical sentences to better explain the role of FUT-2 during rotavirus cell entry (lines 46-47 and 311-312).

---

## [Decision Letter · Decision Letter 1]

27 Jul 2021

Rotavirus susceptibility of antibiotic-treated mice ascribed to diminished expression of interleukin-22

PONE-D-21-04729R1

Dear Dr. Staeheli,

We’re pleased to inform you that your manuscript has been judged scientifically suitable for publication and will be formally accepted for publication once it meets all outstanding technical requirements.

Kind regards,

Michael Nevels

Academic Editor

PLOS ONE

Additional Editor Comments (optional):

Reviewers' comments:

Reviewer's Responses to Questions

**Comments to the Author**

1. If the authors have adequately addressed your comments raised in a previous round of review and you feel that this manuscript is now acceptable for publication, you may indicate that here to bypass the “Comments to the Author” section, enter your conflict of interest statement in the “Confidential to Editor” section, and submit your "Accept" recommendation.

Reviewer #1: (No Response)

Reviewer #2: All comments have been addressed

Reviewer #3: All comments have been addressed

2. Is the manuscript technically sound, and do the data support the conclusions?

Reviewer #1: No

Reviewer #2: Yes

Reviewer #3: Yes

3. Has the statistical analysis been performed appropriately and rigorously? 

Reviewer #1: I Don't Know

Reviewer #2: N/A

Reviewer #3: Yes

4. Have the authors made all data underlying the findings in their manuscript fully available?

Reviewer #1: Yes

Reviewer #2: Yes

Reviewer #3: Yes

5. Is the manuscript presented in an intelligible fashion and written in standard English?

Reviewer #1: Yes

Reviewer #2: Yes

Reviewer #3: Yes

6. Review Comments to the Author

Reviewer #1: While the provide an explanation that they don't mean to declare C57BL/6 with a microbiota un-infectable by RV their data seems to do exactly that. Indeed, this could reflect a low inoculum but, given this work generally lacks novelty and is trying to reconcile some previously published work, they need to at least start with the widely used previous conditions in which C57BL6 mice purchased from standard vendors shed copious amounts of RV. Fine to then use a lower inoculum, but starting with a POSITIVE CONTROL IS ESSENTIAL.

Reviewer #2: In the research article entitled Rotavirus susceptibility of antibiotic-treated mice ascribed to diminished expression of interleukin-22, the authors have incorporated all the suggested changes except the fonts of the figures that are still bold and one can not read them. It is requested to make those fonts readable.

Regards

Reviewer #3: The authors have substantially expanded both the introduction and discussion to put their findings in the context of the field. Additionally, they have clarified experimental details.

7. PLOS authors have the option to publish the peer review history of their article (what does this mean?). If published, this will include your full peer review and any attached files.

Reviewer #1: No

Reviewer #2: **Yes: **Nazish Bostan.

Reviewer #3: No

---

## [Editor Report · Acceptance letter]

4 Aug 2021

PONE-D-21-04729R1 

Rotavirus susceptibility of antibiotic-treated mice ascribed to diminished expression of interleukin-22 

Dear Dr. Staeheli:

I'm pleased to inform you that your manuscript has been deemed suitable for publication in PLOS ONE. Congratulations! Your manuscript is now with our production department. 

Kind regards, 

on behalf of

Dr. Michael Nevels 

Academic Editor

PLOS ONE